# DG-LoRa: Deterministic Group Acknowledgment Transmissions in LoRa Networks for Industrial IoT Applications

**DOI:** 10.3390/s21041444

**Published:** 2021-02-19

**Authors:** Junhee Lee, Young Seog Yoon, Hyun Woo Oh, Kwang Roh Park

**Affiliations:** Industrial IoT Intelligence Research Department, Electronics and Telecommunications Research Institute, Daejeon 34129, Korea; isay@etri.re.kr (Y.S.Y.); hyunwoo@etri.re.kr (H.W.O.); krpark@etri.re.kr (K.R.P.)

**Keywords:** LoRaWAN, GACK, scalability, massive IoT

## Abstract

In this paper, we propose a novel MAC protocol, called DG-LoRa, for improving scalability in low power wide area networks. DG-LoRa is backward compatible with legacy LoRaWAN and adds new features, such as group acknowledgment transmissions in the time-synchronized frame structure that supports determinism on channel access. In DG-LoRa, the number of responses to data frames that are transmitted from end devices is maximized by allocating the spreading factor and timeslot in the frame structure. We evaluate the performance of DG-LoRa using the Monte-Carlo simulation and then compare it with the performance of legacy LoRaWAN in terms of data drop rate and the number of retransmissions. Our numerical results show that DG-LoRa supports approximately five times more connections to the LoRa network satisfying a 5% data drop rate. Also, it is observed that DG-LoRa enables low overhead by reducing the number of data frame retransmissions.

## 1. Introduction

With the increase in the number of industrial internet of things (IoT) applications, IoT connectivity is a critical issue for connecting a massive number of wireless devices to the IoT. Typically, devices that are connected to the IoT are inexpensive, battery-powered for years, while industrial applications need to cover a wide geographical area. These resource constraints of IoT devices make it difficult to support long communication range mainly due to low-cost, low-power operations. A multi-hop network has been introduced so that sensor information originating from a source device is relayed to a destination device in order to overcome the limited communication range. However, as the complexity of network management increases with the use of multi-hop network configurations, there is a large overhead in resource-constrained networks. Low power wide area (LPWA) networks have drawn much attention by offering connectivity over a large area with simple star network configurations. LPWA technologies support low-rate and robust modulation to achieve a multi-kilometer communication range with limited power sources. The market for LPWA technologies is expected to be huge. The Cisco annual internet report states that the number of LPWA connections will increase to 1.9 billion by 2023 [1].

LPWA technologies are characterized by operating in the licensed cellular band and unlicensed band, such as sub- GHz, industrial, scientific, and medical (ISM) band. Long term evolution machine type communication (LTE-MTC), extended coverage global system for mobile communications (EC-GSM), and narrowband IoT (NB-IoT), introduced by the third generation partnership project (3GPP), are LPWA technologies with the licensed cellular band [2]. They reuse the existing cellular infrastructure and radio spectrum, while it is optimized for low-rate IoT applications by reducing the network complexity and operating costs. Meanwhile, LPWA technologies in the unlicensed band are available for organizing user-defined private networks without any cost for frequency use, which results in enabling various industrial IoT applications. In [3], proprietary LPWA technologies that are used in the unlicensed bands are introduced. SIGFOX uses ultra narrow band (UNB) to support wide coverage with low noise levels, and the radio interface is optimized for battery-powered end devices. Ingenu RPMA uses slotted ALOHA protocol that is based on direct sequence spread spectrum (DSSS), and it spreads the desired signal using gold codes. LoRaWAN is a LPWA technology in the sub-GHz unlicensed band using the chirp spread spectrum (CSS), which spreads the desired signal in a narrow band to a wider channel bandwidth by chirp pulses. The spreading technique improves resilience and robustness against wireless interference, as the chirp signal varies its frequency linearly with time. LoRaWAN offers multiple spreading factors (SFs) to decide the tradeoff between communication range and data rate. Higher SF delivers long distance at an expense of lower data rates, while the duration of transmissions is reduced in lower SF with higher data rates. Communications with different data rates in LoRaWAN do not interfere with each other, thus the network infrastructure adjusts data rates for individual wireless links between gateways and end devices to maximize network capacity.

Although LoRaWAN supports the adaptive data rate, the scalability problem has been continuously issued in dense deployments [4,5,6,7,8,9,10]. Half-duplex wireless communication in LoRaWAN is one of the main causes of network scalability problems. In a half-duplex mode, the radio of gateway cannot receive uplink messages that are transmitted from end devices during transmitting downlink messages, thus the uplink messages should be retransmitted. Additionally, the transceiver of the gateway is frequently turned on to send a large number of acknowledgments (ACKs) when the response of uplink messages is required, such as a confirmed type message that is defined in LoRa specification [11]. This implies that the overhead due to the ACK transmission with a large bandwidth should be reduced for connecting a massive number of end devices.

In this paper, we propose DG-LoRa as a MAC protocol that supports deterministic group ACK (GACK) transmissions by aggregating ACKs to improve the scalability of LoRa networks. The contributions of our study consist of the detailed design of DG-LoRa and the strategy of GACK transmissions in a LoRa network with multiple gateways. Firstly, DG-LoRa employs a frame structure in which uplink and downlink transmissions are separated in the time domain to avoid the reception failure that is caused by half-duplex communications. The end device sends a message in the uplink period using simple random channel access, such as ALOHA, and server responses to the received uplink message in the downlink period. The server generates a GACK, including multiple destination device addresses, instead of sending multiple ACKs, in order to reduce the overhead in downlink transmissions. Secondly, DG-LoRa guarantees deterministic GACK transmissions by allocating resources in the time-synchronized frame structure. The server allocates the SF to be used by the gateway for GACK transmission and allocates timeslots to maximize the number of responses. In DG-LoRa, the SF and timeslots are allocated over multiple rounds in the downlink transmission period. These features may maintain the simplicity in channel access of data frames, unlike reservation-based channel access schemes for improving scalability problems in previous studies, which enables simple design and low-cost transceivers of the end device. Additionally, DG-LoRa enhances and adds functionality to legacy LoRaWAN to support the determinism on channel access of GACK transmissions.

The rest of this paper is organized, as follows. In Section 2, previous studies for improving the scalability of LoRa networks are introduced, and we briefly discuss the feature of PHY and MAC layers of LoRaWAN in Section 3. Section 4 describes the detailed design of DG-LoRa, including the frame structure and procedures of GACK transmissions between a server and end devices, and the resource allocation strategy for the GACK transmission is explained in Section 5. In Section 6, we evaluate the performance of DG-LoRa in terms of data drop rate and the number of retransmissions, and then compare it with that of legacy LoRaWAN. Finally, we conclude our study in Section 7.

## 2. Literature Review

The scalability problem of LoRaWAN has been seriously issued as a future challenge to meet the requirements of industrial IoT applications. One of the main reasons for the problem is the ALOHA-based random access for uplink transmissions. In the random access, end devices send information without doing any carrier sensing. This simplicity in channel access enables simple design and low-cost transceiver. However, random behavior makes it difficult to scale the network due to the mutual interference with other transmissions on the channel.

Several resource allocation methods for accessing the medium channel have been investigated to overcome the scalability problem due to random access. In [12], a centralized resource allocation mechanism is proposed, in which the network server allocates communication resources for entire link transmissions in a LoRa network. An end device sends a message indicating a synchronization request to the network server, and the network server returns the timeslot allocation information for communication between the gateway and end device. They employ a probabilistic data structure using the Bloom filter to support low overhead in reply message transmissions. The study in [13] introduces a MAC protocol providing two-step lightweight scheduling in a distributed manner to improve the scalability and reliability of LoRaWAN. The gateway transmits a beacon frame containing coarse-grained information of SF and allowed transmission power in the first stage. In the second stage, the end device determines its own transmission parameters, such as frequency channel, SF, and transmission power, by referring to the parameters in the received beacon frame. These assignment mechanisms can alleviate the capture effect and the probability of packet collision due to random access in uplink transmissions.

Despite the advantages of allocating resources, the random access approach has been considered due to its simplicity. In [14,15], the performance of LoRaWAN enabling carrier-sense multiple access (CSMA), end devices listen to the channel before attempting transmissions, is evaluated. They show the performance of LoRaWAN with CSMA improves than that of the ALOHA-based network through mathematical modeling and network simulation. The study in [16] proposes a MAC protocol utilizing a frame structure consisting of a contention period, an uplink transmission period, and a downlink transmission period. In this protocol, a device transmits a control message for a channel access request in a contention period before attempting uplink transmission. A device that successfully delivers the request message sends a data frame in the uplink transmission period, and the gateway broadcasts the feedback information to execute the rules of protocol in a distributed manner.

There have been significant research efforts on GACK transmissions to improve the scalability of LoRaWAN. In [17,18], MAC protocols supporting multiple ACKs with a single downlink transmission on the time-slotted frame structure are introduced. The server aggregates ACKs to respond to the end device sending an uplink message to the gateway and returns a GACK to the end device. In the studies, they do not consider SF allocations for GACK transmissions. This implies a long time duration for the GACK transmission is required, since the higher SF should be used so that the GACK can be delivered to the destination device farthest from the gateway. In [19,20], GACK transmission mechanisms using a time-slotted periodic frame structure that consists of several subframes classified by SF are introduced. This transmission strategy can reduce the duration of GACK transmission by aggregating ACKs for uplink transmissions with the same SF, but the mechanisms do not guarantee the determinism on channel access of GACK transmissions.

## 3. LoRa Overview

The LoRa network is organized in a star-of-stars configuration, in which gateways relay the messages between end devices and a network server. Gateways are connected to the network server via wired IP connections, while single-hop wireless links are dedicated to communications between end devices and gateways. Communications over the wireless links are spread out by exploiting the CSS technique on the physical layer of LoRaWAN.

The spreading of the desired signal is achieved by generating a chirp signal that consists of $2^{SF}$ chirps in a LoRa symbol. Higher SF enables a lower data rate by increasing the number of chirps to spread a signal. The data rate $r_{b}$ is given, as follows:(1)$$r_{b}=w/2^{SF}$$
where *w* denotes the receiver bandwidth. Also, $r_{b}$ decides the minimum ratio of desired signal power to noise that can be demodulated. The ratio, denoted by σ, is given. as follows:(2)$$\sigma =E_{b}/N_{o}*r_{b}/w$$
where $E_{b}/N_{o}$ denotes the energy per bit to noise power spectral density ratio. Because the sensitivity of the receiver increases as the σ decreases, higher SF allows for wide coverage due to the higher sensitivity of the receiver. Let us denote that the receiver sensitivity is $s_{r}$. Subsequently, the relationship between the $s_{r}$ and σ is expressed, as follows:(3)$$s_{r}=-174+10\log_{10}w+n+\sigma$$
where *n* is the noise figure. However, higher SF leads to an increase in energy consumed for message transmissions. This mainly because symbol duration increases by $2^{SF}$ times, as SF increases. The duration of a message transmission $T_{p}$ is given, as follows:(4)$$T_{p}=T_{s}\left(n_{p}+4.25+w_{s}+\max\left(\gamma ,0\right)\right)$$
(5)$$\gamma =\frac{8l-4SF+28+16-20h}{4(SF-2e)}\left(r_{c}+4\right)$$
where $n_{p}$ is the number of preamble symbols, $w_{s}$ is the synchronization words in symbols, and $T_{s}$ denotes the symbol duration, which is $2^{SF}/w$. In (Equation 5), *l* is the payload size in bytes and $r_{c}$ is the coding rate, with $r_{c}$∈ {1, 2, 3, 4}. The *h* indicates the presence of the physical header, and the use of data rate optimization is described by the *e* (1: enabled, 0: disabled). Table 1 shows the influence of SF on the receiver sensitivity and duration of transmission under the conditions; *w* = 125 kHz, $r_{c}$ = 1, *l* = 10, *e* = 0, and *h* = 0. When the SF increases by one, the sensitivity increases by 3dB, but the transmission length increases by two times. Consequently, a trade-off between communication range and transmission duration is required.

LoRaWAN defines three different types of end devices depending on the capabilities and requirements of power consumption. Class-A device consumes the least amount of energy among all types of end devices in LoRaWAN. It turns off the radio frequency (RF) transceiver until a message to be sent is generated, and sends the message by turning on the transceiver if the message is generated. After sending the message, the end device opens two receive windows to receive downlink transmissions. This transmission mechanism allows low power operation, but the end device cannot receive other downlink messages until the next uplink message is sent. On the other hand, the Class-B device opens scheduled several receive windows for enabling network-initiated downlink transmissions. The end device is time-synchronized by a periodic beacon sent from gateways to schedule downlink transmissions without any prior uplink transmissions. Class-C device supports flexibility in downlink transmissions by continuously opening receive windows, but they consume a large amount of energy due to listening to the channel for a long time. Figure 1 describes the transmission and reception slot timing of three different types of end devices.

## 4. DG-LoRa Design

This section first introduces the time-synchronized frame structure of DG-LoRa to guarantee deterministic GACK transmission. Subsequently, the detailed procedure of GACK transmission between a gateway and end devices is described.

### 4.1. Frame Structure

DG-LoRa uses a time-synchronized frame structure with a periodic beacon described in Figure 2. The first portion of the frame structure is the beacon period (BP). In the BP, the gateway sends a beacon at the beginning of the frame structure using SF allocated by the network server. The allocation enables multiple beacon transmissions by allowing each gateway to transmit beacons while using a different SF. After the beacon transmission, an end device establishes its frame structure by receiving the beacon from the gateway. The portion of the frame following the BP is called the uplink transmission period (UTP). In the UTP, the end device sends a message using pure ALOHA, which is a MAC protocol of LoRaWAN. If the uplink message is a confirmed type to be acknowledged by the gateway, the response message is sent in the downlink transmission period (DTP) following UTP. DG-LoRa allocates SF for GACK transmission under the premise that the server knows SF used for data transmission according to an SF allocation scheme for uplink transmission. In DTP, the server allocates resources to reply to the device with the same SF as the SF used for data frames that are sent from the device. Subsequently, multiple responses that can allocate the same SF are aggregated to send GACK. This strategy allows for the end device to receive GACKs without exchanging control messages with the server.

The DTP is divided into two dimensions that cosnist of both SF and timeslot (TS). The GACK transmission using the minimum SF (i.e., SF7) supported by LoRaWAN is available by allocating one timeslot in the DTP. On the other hand, GACK transmission using SF larger than seven consumes multiple timeslots due to an increase in transmission duration. Because the duration of transmission increases twice every SF increases, two times the number of timeslots are consumed as SF increases by one.

Figure 3 describes an example of timeslot allocation for GACK transmission in a network environment where SFs from SF7 to SF10 are available for communications between four gateways and end devices. We assume that $G_{j}$ is the GACK transmission from gateway j, being denoted by $g_{j}$. First of all, each gateway transmits a GACK using different SFs at the beginning of the DTP. Among all gateways sending GACKs in the first timeslot, $g_{1}$ ends GACK transmission at the earliest time using the lowest SF for the transmission. After $g_{1}$ completes the GACK transmission in the first timeslot, it sends a newly generated GACK using SF7, which is not used to transmit the GACKs from other gateways in the second timeslot. In the third timeslot, $g_{1}$ and $g_{2}$ send other GACKs using the remaining SFs, except for the SFs that were used by GACK transmissions from $g_{3}$ and $g_{4}$. These procedures may acknowledge receipt of a large number of uplink transmissions by simultaneously transmitting multiple GACKs using different SFs. If some data frames transmitted from end devices are not acknowledged due to the limited number of GACK transmissions in the current subframe, a GACK for the end devices generates in the next subframe.

### 4.2. GACK Transmission Procedures

A GACK includes *NumDev* and *gaddr* fields indicating the number of destination devices and the address of the end devices. The *gaddr* field is dynamically generated according to the number of destination devices, and each field contains an address of the end device in a size of 4 bytes. The number of *gaddr* fields can be increased within the maximum MAC payload size rhat was determined by the data rate used for GACK transmissions. This implies that the maximum number of destination devices that can be included in the GACK relies on which SF is used to transmit the GACK. Table 2 describes the maximum number of *gaddr* fields in the GACK according to SF with rhw 125 KHz channel bandwidth.

The end device retrieves its address from the *gaddr* field of the transmitted GACK using the same SF used to transmit the uplink message. If the address of the end device is included in the GACK, the end device confirms the successful delivery of its data frame. Otherwise, the end device may check another GACK to be delivered in the next timeslot in order to confirm the successful data transmission. Meanwhile, some end devices may not be able to receive the GACK in the current DTP, since the number of GACK transmitted in the DTP is not sufficient for including the address of all end devices transmitting data frames. In this case, the end device retransmits its uplink message in the next subframe.

Figure 4 describes the procedure for exchanging GACK to acknowledge data frames that are transmitted from four end devices. In this figure, the end device with the address *i* is denoted by $d_{i}$, and the uplink transmission from $d_{i}$ is represented by $u_{i}$. We assume that SF7 is used for transmitting GACKs to be delivered to $d_{1}$ and $d_{4}$, while SF8 is considered for communications between gateways and both $d_{2}$ and $d_{3}$ end devices. In this case, multiple GACKs can be simultaneously transmitted via two gateways using different SFs. The destination devices of GACKs that are transmitted from $g_{1}$ are $d_{1}$ and $d_{4}$, and $g_{1}$ sends the GACK by using SF7 in the first timeslot of DTP. While another gateway $g_{2}$ sends GACK to $d_{2}$ and $d_{3}$ end devices by allocating the first and second timeslots.

## 5. Resource Allocations for DG-LoRa

The main concern of DG-LoRa is to allocate SF and timeslots to maximize the number of responses via GACK transmission. In this section, we present a mathematical formulation for maximizing the number of responses and details the resource allocation algorithm in order to achieve the objective.

### 5.1. Problem Formulations

DG-LoRa allocates SFs $\theta^{*}$ and timeslots $\tau^{*}$ for all GACKs to be transmitted in the DTP to maximize the number of responses to data frames transmitted in the UTP of the subframe. We denote the set of *M* gateways as *I* = {1,...,*M*}, the set of *N* end devices as *J* = {1,...,*N*}, the set of timeslots of the DTP as *K*, and the set of SFs to be used for sending GACKs is denoted as *S*, in order to explain the problem formulation attributed to the objective of DG-LoRa. We define a binary variable $r_{ijk}$, where $r_{ijk}$ is 1 if gateway *i* starts sending a GACK to end device *j* in the *k*th timeslot of DTP, and $r_{ijk}$ is 0 if otherwise. We also define $s_{ik}$ as the SF that gateway *i* uses to send GACKs in the *k*th timeslot of the DTP. Now, the mathematical formulation of our study is represented, as follows:(6)$$\max_{\theta^{*},\tau^{*}}\sum_{i\in I}\sum_{j\in J}\sum_{k\in K}r_{ijk}$$
subject to
(7)$$\sum_{i\in I}\sum_{k\in K}r_{ijk}\leq 1,\;\forall j\in J$$
(8)$$\sum_{k\in K}r_{ijk}\leq 1,\;\forall i\in I,\forall j\in J$$
(9)$$\sum_{j\in J}r_{ijk}\leq \pi \left(s_{ik}\right),\;\forall i\in I,\forall k\in K$$
(10)$$s_{ik}\neq s_{\hat{i}k},\;\forall i\in I,\hat{i}\in I\backslash \left\{i\right\},\forall k\in K$$
(11)$$s_{ik}\in S,\;\forall i\in I,\forall k\in K$$
where $\pi (s_{ik})$ is the maximum number of end devices that can be included in a GACK sent using $s_{ik}$. The constraint in (Equation 7) indicates that the destination device address of the GACK transmitted in a given DTP cannot be duplicated. The constraint presented in (Equation 8) means that the number of destination devices that are included in the GACK allocated to SF of $s_{ik}$ cannot exceed $\pi (s_{ik})$. The constraint in (Equation 9) ensures that multiple GACKs sent simultaneously from gateways use different SFs.

### 5.2. Resource Allocation Algorithm

DG-LoRa allocates SF and timeslots for GACK transmission over multiple rounds. DG-LoRa sequentially increases the timeslot index from the first timeslot of the DTP to the last timeslot to check whether there is an idle gateway, which refers to a gateway that does not transmit GACK in the current timeslot. Note that the gateway becomes idle when it completes GACK transmission in the previous timeslot. Additionally, the gateway that has never been allocated resources for GACK transmission in DTP is also defined as an idle gateway. The server defines the current timeslot by increasing the timeslot index, and it found an idle gateway in the current timeslot. If an idle gateway is found in the process of increasing the timeslot index, a new round is started and resources are allocated, so that the idle gateway can start transmitting GACK in that timeslot.

DG-LoRa allocates resources using information from a set of devices that can be included in GACK according to the SF that is used by the gateway to transmit GACK. Let us denote that U(i,s) is the set of end devices that can receive GACK when gateway *i* uses SF *s* for GACK transmission, where i∈I and s∈S. Subsequently, DG-LoRa refers to *U* in order to determine the SF that is to be used by each gateway for GACK transmission to achieve the objective of maximizing the number of end devices receiving GACK. To describe the resource allocation procedure referring to *U*, we define a *m*-tuple, denoted as $\theta_{r}=\left(a_{1},\ldots ,a_{m}\right)$, representing the SF used for GACK transmission by *m* gateways in the round *r*. For example, $\theta_{r}$ = (7,8,9,10) indicates that the first gateway uses SF7, the second one uses SF8, and the remaining gateways use SF9 and SF10 for GACK transmission. If some gateways are not idle, then the element value $\theta_{r}$ corresponding to the gateway is set to zero. Now, the number of end devices that can be included in the GACK when gateway *i* uses $\theta_{r}\left(i\right)$ is given, as follows:(12)$$N_{i,\theta_{r}\left(i\right)}=min\left(n\left(U(i,\theta_{r}\left(i\right))\right),L\left(\theta_{r}\left(i\right)\right)\right)$$
where $\theta_{r}\left(i\right)$ is the *i*th element of $\theta_{r}$, and $L(\theta_{r}\left(i\right))$ is the maximum number of end devices that can be included in the GACK when SF used for GACK transmission is $\theta_{r}\left(i\right)$. For a given $\theta_{r}$, the number of end devices included in the GACK transmitted in the round, denoted by $N_{\theta }^{r}$, is described, as follows:(13)$$N_{\theta }^{r}=\sum_{i\in I}N_{i,\theta_{r}\left(i\right)}$$

DG-LoRa searches for a $\theta_{r}^{*}$ achieving the objective among all $\theta_{r}$ that can be generated every round and allocates SF that is defined in $\theta_{r}^{*}$ for GACK transmission. Based on the $\theta_{r}^{*}$, timeslots are allocated by referring to SFs that are defined in $\theta_{r}^{*}$. We define a two-tuple denoted as $\tau_{r}\left(i\right)=\left(t,t^{{}^{\prime }}\right)$ representing the first timeslot *t* and the last timeslot $t^{{}^{\prime }}$ allocated for GACK to be transmitted by gateway *i* in the round *r*. For example, $\tau_{r}\left(i\right)=\left(t,t+1\right)$ if $\theta_{r}\left(i\right)=8$, since two timeslots are required when SF8 is used for the GACK transmission. After resource allocation for GACK transmission in the round is completed, end devices included in GACK are removed from *U*, and the resources are allocated by referring to the updated *U* in the next round. Based on the above explanation, the GACK transmission algorithm of DG-LoRa is described as Algorithm 1.
**Algorithm 1.** Resource Allocation Algorithm for GACK Transmission**Input**: *U***Output**: $\theta_{r}^{*}$, $\tau_{r}^{*}$, ∀*r***Initialization**: $N_{\theta }^{*}$← 0, *r* ← 11: **for**
*t* ← 1 to *k*2:    **if** there is an idle gateway on the timeslot *t*3:       Generate $\theta_{r}$ and compute $N_{\theta }^{r}$4:       **if**
$N_{\theta }^{r}$ >${N_{\theta }^{r}}^{*}$
**then**5:          ${N_{\theta }^{r}}^{*}$ ← $N_{\theta }^{r}$, $\theta_{r}^{*}$ ← $\theta_{r}$6:          $\tau_{r}^{*}\left(i\right)$ ← $\left(t,t+\theta_{r}^{*}\left(i\right)-min\left(S\right)\right)$, ∀*i*7:       **end if**8:      Repeats 3-9 until all $\theta_{r}$ are considered9:   **end if**10:   *r* ← *r*+111:   Update *U*12:**end for**

Figure 5 describes the resource allocation procedure for GACK transmission in the network environment introduced to explain Figure 3. Because all gateways are idle gateways in the first timeslot of DTP, resources are allocated, so that multiple GACKs from all gateways are simultaneously transmitted using different SFs (e.g., $\theta_{1}^{*}=\left(7,8,9,10\right)$). The $g_{1}$ in which GACK transmission allocated in the first round is completed first becomes an idle gateway in the second timeslot, and resources are allocated in the second round, so that the $g_{1}$ can start GACK transmission in the second timeslot. In the second timeslot, all of the SFs, except SF7, are already occupied by other GACK transmissions, thus resources are allocated in the second round, so that the $g_{1}$ gateway sends GACK using SF7. In the third round, resources are allocated, so that gateways $g_{1}$ and $g_{2}$, which have completed GACK transmission allocated in the previous two rounds, can start GACK transmission in the third timeslot. Assume that the number of end devices receiving GACK is maximized when using SF8 and SF7 for GACKs that are sent by $g_{1}$ and $g_{2}$ gateways (i.e., $\theta_{3}^{*}=\left(8,7,0,0\right)$). Through resource allocation according to this round repetition, SF and timeslots are allocated, so that idle gateways of each round can transmit GACK until the end of the DTP section.

## 6. Performance Evaluation

In this section, we evaluate the performanceof DG-LoRa using Monte-Carlo simulation and compare it with the performance of legacy LoRaWAN. End devices are randomly deployed within a geographical area with a radius of *l* kilometers, which is the maximum distance the device can transmit a data frame using SF12 based on log-distance path loss model used in LoRaSIM [21], described as follows:(14)$$L_{pl}\left(l^{{}^{\prime }}\right)=\bar{L_{pl}\left(l_{0}\right)}+10\gamma \log\left(l^{{}^{\prime }}/l_{0}\right)+X_{\sigma }$$
where $L_{pl}\left(l^{{}^{\prime }}\right)$ is the path loss in dB at the distance $l^{{}^{\prime }}$ between a gateway and an end device, $\bar{L_{pl}\left(l_{0}\right)}$ is the mean path loss at the reference distance $l_{0}$, γ is the path loss exponent, and $X_{\sigma }$ is the normal distribution, with zero mean and $\sigma^{2}$ variance. The simulation scenario takes the geographical area of approximately 947 meters in the radius into account when substituting the simulation parameters that are described in Table 3 into Equation (Equation 14). Gateways are located in a fixed position according to the number of gateways, and from 500 to 5000 end devices are randomly deployed in the geographical area, as shown in Figure 6. In the deployment with 500 end devices, approximately 178 end devices per km${}^{2}$ are dense in the area, while the density is 1775 end devices per km${}^{2}$ in the deployment with 5000 end devices. The SF, frequency channel and bandwidth used for transmissions on the wireless links connecting the gateways and the end devices refer to the parameters of the US 902–928 MHz ISM band that is defined by LoRa specification. SF used for the uplink transmission is chosen randomly from all possible SFs that enable communication between the gateway and the end device, and the selected SF is also used for the GACK transmission by setting the *RX1DROffset* to zero. All of the frequency channels are available for each uplink transmissions, while a common frequency channel is used for GACK transmissions. In the configuration, each end device periodically generates a data frame for every BI and sends the data to the gateway via ALOHA-based random access. Table 3 lists the simulation parameters, and the remaining parameters that are not expressed in Table 3 refer to the LoRa specification [11].

We employ the data drop rate (DDR) as a performance metric by varying the number of end devices to evaluate how many end devices can be connected to the LoRa network. The performance is evaluated in a star-of-star network topology, which consists of multiple end devices and one or more gateways. Additionally, we investigate how many retransmissions occur to successfully deliver a data frame. It should be noted that the amount of retransmissions is a crucial metric for evaluating the performance of DG-LoRa in resource-constrained wireless networks, such as LoRa networks.

### 6.1. Data Drop Rate Analysis

We evaluate the DDR against the number of end devices to investigate how many end devices can be connected to the network. The DDR represents the ratio of the number of dropped data frames to the total number of arrival data frames. The dropped data frame indicates that the frame cannot be successfully delivered after MaxNumRetransmission transmission attempts, where MaxNumRetransmission is the maximum allowable number of retransmissions. The MAC protocols DG-LoRa and LoRaWAN that are to be evaluated the performance are inherently subject to transmission failure, due to ALOHA-based random access for data transmission. However, there are differences between the two protocols in the process of responding to data transmissions. In LoRaWAN, other transmissions may interfere with an ACK transmission, while DG-LoRa supporting deterministic GACK transmission does not cause transmission failure by colliding with other transmissions in the network. However, GACK transmission failure may occur due to the limitation of the number of GACKs that can be transmitted in DTP. If the transmission failure occurs, even after MaxNumRetransmission transmission attempts due to such transmission failure factors, the data frame is dropped. In this section, we evaluate DDR performance and discuss the results that are based on the difference in the process of responding to data transmissions between legacy LoRaWAN and DG-LoRa.

Figure 7 illustrates the DDR of both LoRaWAN and DG-LoRa as the number of end devices increases. In the figure, we represent the performance behavior against different channel bandwidth of 125 KHz and 500 KHz. In this scenario, two gateways are connected to the LoRa network, and the coding rate sets to one. It is observed that the DDR of LoRaWAN and DG-LoRa increases as the number of end devices that are deployed in the network increases. The degradation of performance due to the increase in the number of collisions of transmitting messages as the number of end devices increases. However, the performance of DG-LoRa is higher than that of LoRaWAN, regardless of the number of end devices that are connected to the LoRa network. The simulation results show that DG-LoRa connects approximately five times more end devices to the network than legacy LoRaWAN, while satisfying 5% DDR. DG-LoRa supports over 2500 end devices to be connected to the LoRa network with 125 KHz channel bandwidth if the target DDR is 5%, while LoRaWAN is difficult to connect even 500 end devices to meet the target performance. This is because DG-LoRa improves network scalability by supporting GACK transmission in a frame structure that separates the uplink and downlink periods. Meanwhile, the performance with 500 KHz channel bandwidth slightly increases compared to that of 125 KHz channel bandwidth. This implies that a higher data rate in 500 KHz channel bandwidth is the dominant factor in performance improvements.

Figure 8 shows the performance behavior of the DDR according to the number of gateways to investigate how many gateways are required to connect end devices that are deployed in the geographical area. We consider both single gateway and multi-gateway environments in this scenario. In single gateway environments, one gateway that is located in the center of the area is connected to the network, while several gateways are located on the fixed position in the multi-gateway environment, as shown in Figure 6. The simulation results show that DG-LoRa and LoRaWAN are difficult to achieve less than 5% DDR when more than 1000 end devices are deployed with a single gateway environment. This means that the single gateway makes it difficult to communicate over wireless links with thousands of end devices due to its half-duplex RF transceivers. Meanwhile, increasing the number of gateways improves the network performance. DG-LoRa supports over 5000 connections while satisfying a 5% DDR in the LoRa network with four gateways, and LoRaWAN can connect 1000 end devices to the network. This is because the number of GACKs transmitted by gateways in the DTP section increases, thus more devices can receive responses to data transmissions.

Figure 9 describes the DDR against different coding rates. As expressed in Equations (Equation 4) and (Equation 5), the coding rate directly affects the duration of message transmission, thus the performance behavior due to the variation of frame length can be observed in this scenario. We consider two gateways in the LoRa network with a channel bandwidth of 500 KHz. Simulation results show that network performance improves as the coding rate decreases, but the performance difference is not noticeable as compared to the gap of performance according to the number of gateways and channel bandwidth. However, it is observed that the difference in performance of LoRaWAN is slightly larger than that of DG-LoRa. In a network environment where 5000 devices are deployed, the performance gap between different coding rates is approximately 20%, which is higher than the performance difference in DG-LoRa. This is mainly because the number of other transmissions interrupted by the long duration of transmission in LoRaWAN is larger than that of DG-LoRa. In LoRaWAN, the transmission interferes with other transmissions while using the same frequency channel and SF regardless of the direction of link transmissions. On the other hand, DG-LoRa supports a frame structure in which uplink and downlink transmissions are separated to avoid interference between uplink and downlink transmissions, thus the performance degradation is not noticeable by increasing coding rates.

### 6.2. Retransmission Analysis

Retransmissions reduce the efficiency of communications in resource-constrained wireless networks due to long channel occupancy times for a data frame transmission. Especially, in LoRa networks with long frame lengths with low-rates, the retransmission may interfere with other transmissions, which results in significant overhead. In this section, we evaluate the retransmission performance of DG-LoRa and legacy LoRaWAN in order to investigate how much overhead is incurred depending on the network environment. As a performance metric, we define normalized the number of retransmissions by dividing the average number of retransmissions for a successful data frame transmission by MaxNumRetransmission. Let us denote that normalized the number of retransmissions is δ, and δ is expressed, as follows:(15)$$\delta =\left(\frac{1}{n}\sum_{q\in Q}\rho_{q}\right)/\beta ,\;\rho_{q}\in \left\{1,2,\ldots ,\beta \right\}$$
where $\rho_{q}$ is the number of retransmissions to deliver the *q* data frame successfully, *Q* is the set of all the data frames transmitted by end devices, β is MaxNumRetransmission, and *n* is the number of arrival data frames. The value of ρ being close to zero means that retransmission rarely occurs, whereas the value of ρ approaches to one indicates that the number of retransmissions close to MaxNumRetransmission is required for the data transmission.

Figure 10 illustrates the normalized number of retransmissions of both LoRaWAN and DG-LoRa against different channel bandwidth. In order to observe the performance for the different channel bandwidth, 125 KHz and 500 KHz bandwidths in the US 902–928 MHz band are considered. It is observed that the number of retransmissions for both protocols with a channel bandwidth of 500 KHz is less than that of 125 KHz channel bandwidth. The higher data rate used in the 500 KHz channel bandwidth reduces the overhead in the LoRa network by reducing the number of retransmissions. Especially, the performance difference according to the channel bandwidth of LoRaWAN is noticeable. The simulation results show that the performance difference between the two channel bandwidths by up to 2.6 times in LoRaWAN, which is larger than that of DG-LoRa. The difference in LoRaWAN is because transmitting at the higher data rate may reduce the radio interference with other uplink and downlink transmissions. On the other hand, the variation of rate in the data frame does not affect downlink transmissions due to the determinism of the channel access of the downlink transmissions. This means that the variation of data rate in DG-LoRa has less effect on other transmissions than LoRaWAN, thus the difference in the performance of DG-LoRa according to the channel bandwidth is not noticeable. Meanwhile, the performance difference between the two protocols is observed at a given channel bandwidth. In the simulation results, the normalized number of retransmissions of LoRaWAN grows by more than 0.9 when 5000 end devices are deployed in the network, while that of DG-LoRa is less than 0.3. Becauses the frame structure of DG-LoRa provides deterministic GACK transmissions, the retransmission overhead in the LoRa network can be reduced by employing DG-LoRa.

Figure 11 shows the performance behavior of the number of retransmissions according to the number of gateways, which forward frames between the network server and end devices. The location of gateways is the same as the deployment that was considered in the DDR analysis. In this scenario, a 500 KHz channel bandwidth is used, and the coding rate sets to 1. It is observed that increasing the number of gateways reduces the number of retransmissions of data frames. In LoRaWAN, the normalized number of retransmissions is approximately 0.9 in the single gateway environment, and the performance in the multiple gateway environment is more than doubled when compared to the network with a single gateway. On the other hand, the performance of DG-LoRa with a single gateway is similar to that of LoRaWAN with four gateways, and DG-LoRa supports 5000 connections with the normalized number of retransmissions of 0.05 by increasing deployments of gateways. This difference in the number of retransmissions is mainly because the operation by the single gateway makes it difficult to exchange a large number of frames being exchanged between the network server and end devices. By increasing the deployments of gateways, the overhead due to retransmissions can be reduced by increasing the capacity for frame exchange.

Figure 12 illustrates the performance behavior according to the coding rate in order to investigate the effect of frame length on the retransmissions. In this scenario, we employ a 500 KHz channel bandwidth for frame exchanges in the network with two gateways. The simulation results show that the number of retransmissions is slightly reduced as the coding rate decreases. In LoRaWAN, the normalized number of retransmissions is observed up to 0.8 when the coding rate is 4, and even when the coding rate is lowered to 1, the performance is more than 0.7. While the performance of DG-LoRa is less than 0.3 for all coding rates. The performance variation with coding rates is less than that of LoRaWAN due to the deterministic GACK transmission characteristics of DG-LoRa. The difference in the number of retransmissions between both protocols increases as the number of end devices increases. This implies that DG-LoRa supports a large number of connections by consuming less energy than LoRaWAN. This implies that DG-LoRa supports a large number of connections by reducing the retransmission overhead than LoRaWAN.

## 7. Conclusions

This paper proposed DG-LoRa, a MAC protocol that provides deterministic GACK transmissions for LoRa networks. The frame structure for DG-LoRa was introduced, and we presented the SF and timeslot allocation strategies for the GACK transmission. This paper discussed the performance behavior of DG-LoRa and compares it with the performance of legacy LoRaWAN in terms of DDR and the number of retransmissions. The main difference between the two protocols is that DG-LoRa supports GACK transmissions by aggregating ACKs with reservation-based channel access, while the response of the data frame in the legacy LoRaWAN is individually transmitted after a certain delay time. The performance is evaluated by varying the number of gateways, channel bandwidth, and coding rate. The simulation results show that DG-LoRa improved the performance of the legacy LoRaWAN in terms of DDR and the number of retransmissions. DG-LoRa connects approximately five times more end devices to the LoRa network than legacy LoRaWAN while satisfying 5% DDR. In DG-LoRa, more than 2500 end devices can be connected to the LoRa network with 125 KHz channel bandwidth if the target DDR is 5%, while it is even difficult for LoRaWAN to connect 500 end devices to meet the target performance. In addition, DG-LoRa enables low-power operation of end devices by reducing the number of retransmissions by more than three times when compared to legacy LoRaWAN. In addition, DG-LoRa enables low overhead by reducing the number of retransmissions by more than three times as compared to legacy LoRaWAN. In the network with 5000 end devices deployment, the normalized number of retransmissions of LoRaWAN grows by more than 0.9, while that of DG-LoRa is less than 0.3. These findings mean that DG-LoRa is a suitable protocol for resource-constrained massive IoT networks. Our future research direction is to study MAC protocol for improving the reliability of low power wide area networks. It is expected that network reliability can be improved by applying a resource allocation scheme for uplink transmission along with deterministic GACK transmission of DG-LoRa.

## Figures and Tables

**Figure 1 sensors-21-01444-f001:**
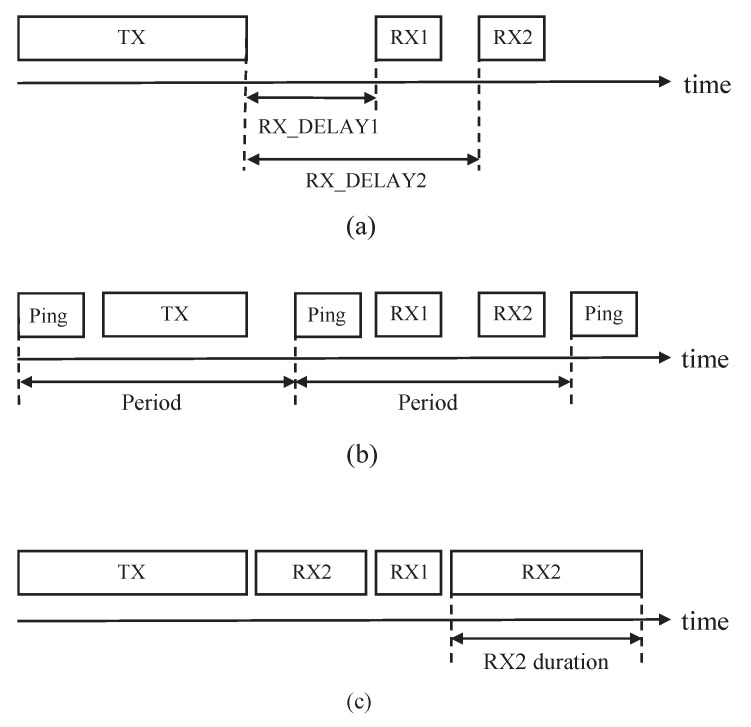
Transmission and reception slot timing of the end device in LoRaWAN. (**a**) Class-A. (**b**) Class-B. (**c**) Class-C.

**Figure 2 sensors-21-01444-f002:**
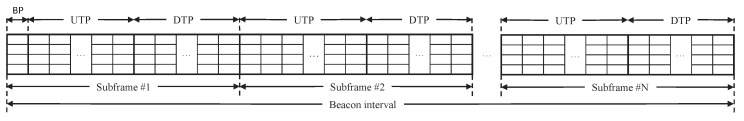
A frame structure for DG-LoRa.

**Figure 3 sensors-21-01444-f003:**
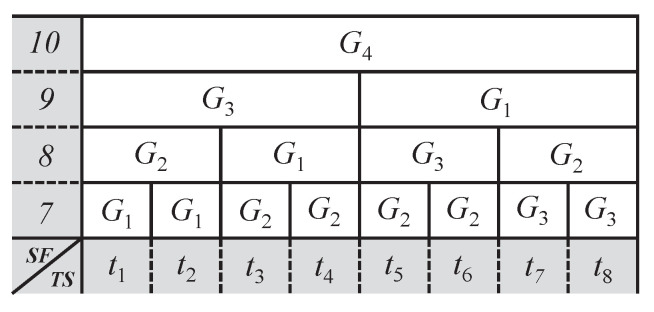
An example of resource allocations for GACK transmissions.

**Figure 4 sensors-21-01444-f004:**
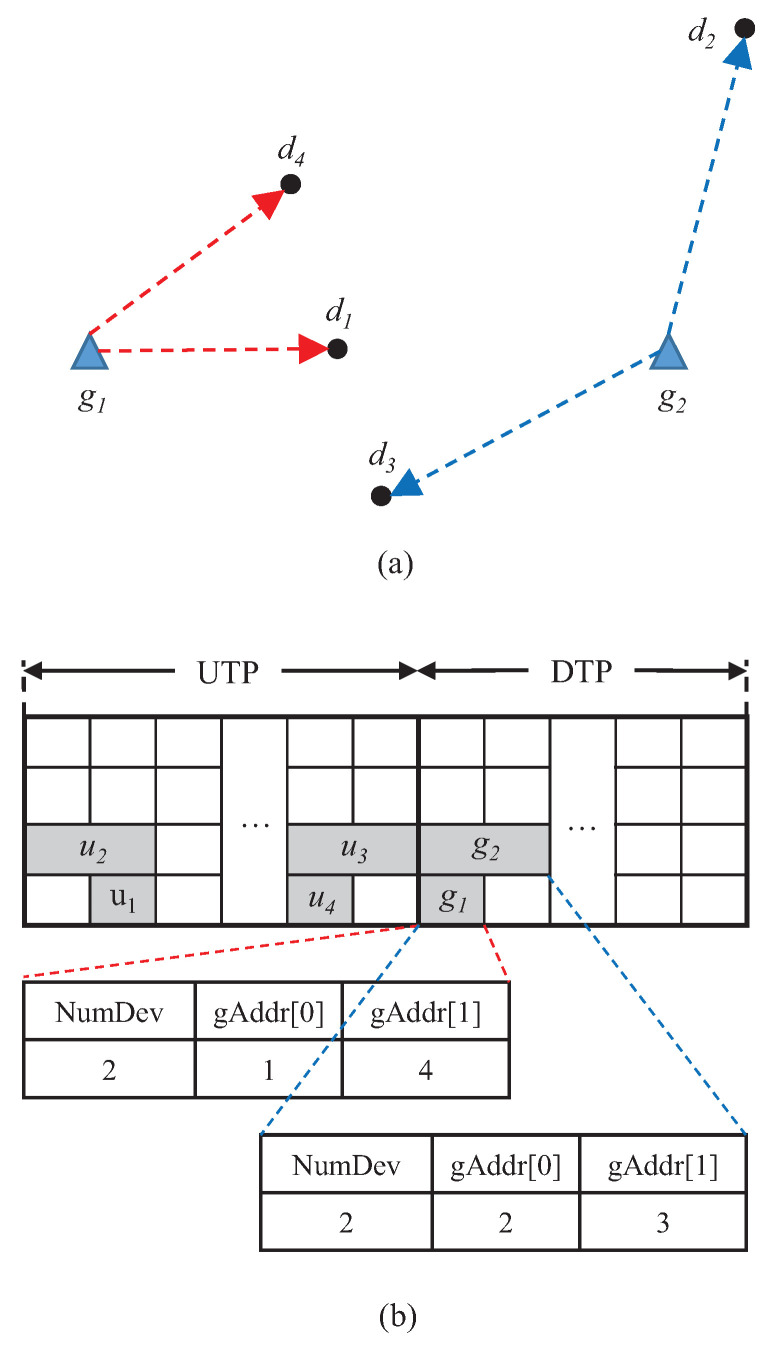
Illustration of exchanging GACKs in the network with multiple gateways. (**a**) Multiple GACK transmissions from two gateways. (**b**) Description of *NumDev* and *gaddr* fields in GACKs.

**Figure 5 sensors-21-01444-f005:**
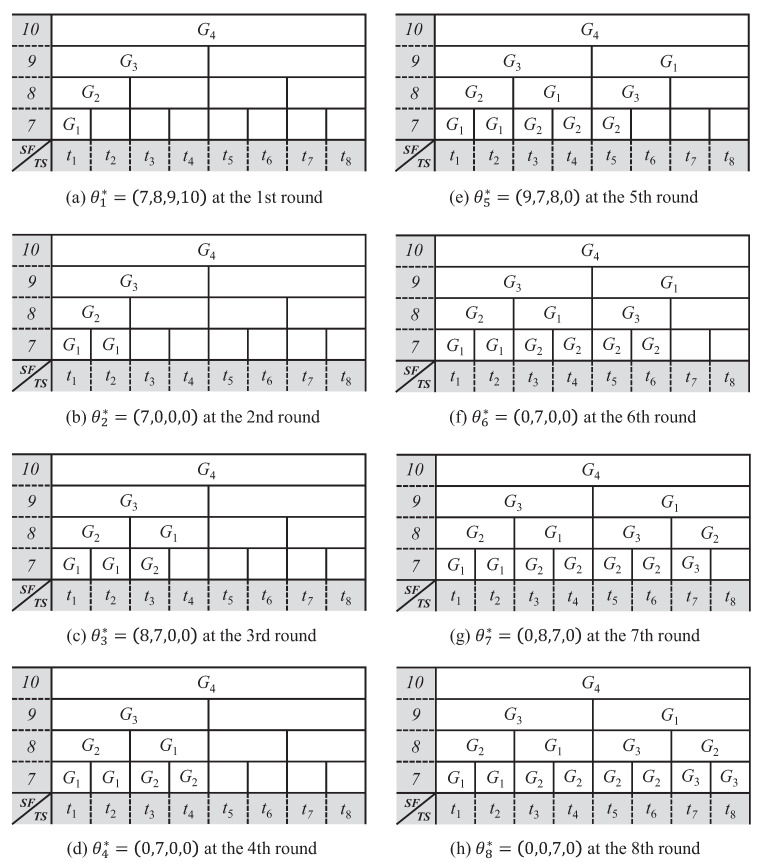
Procedures of allocation in SF and timeslot for GACK transmission in downlink transmission period (DTP).

**Figure 6 sensors-21-01444-f006:**
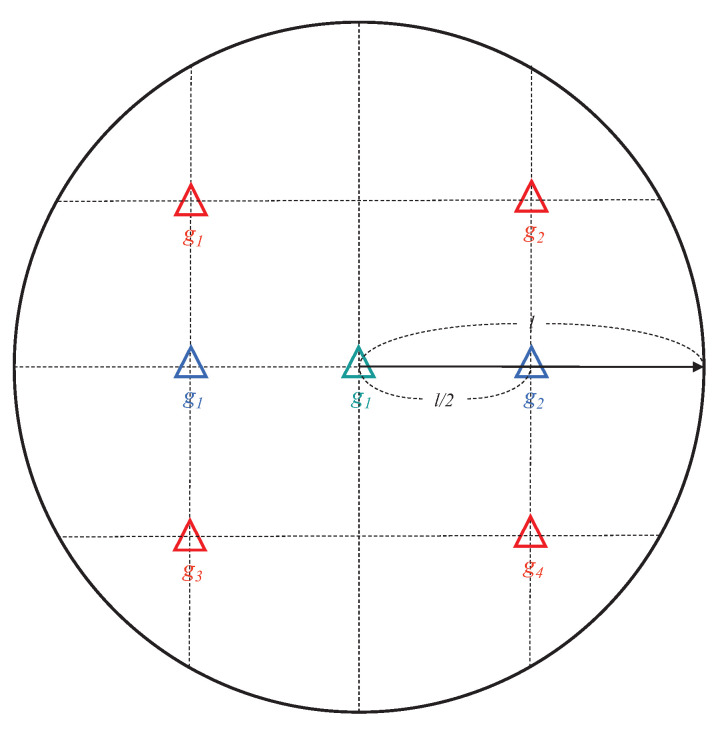
Deployments of gateways for network simulation.

**Figure 7 sensors-21-01444-f007:**
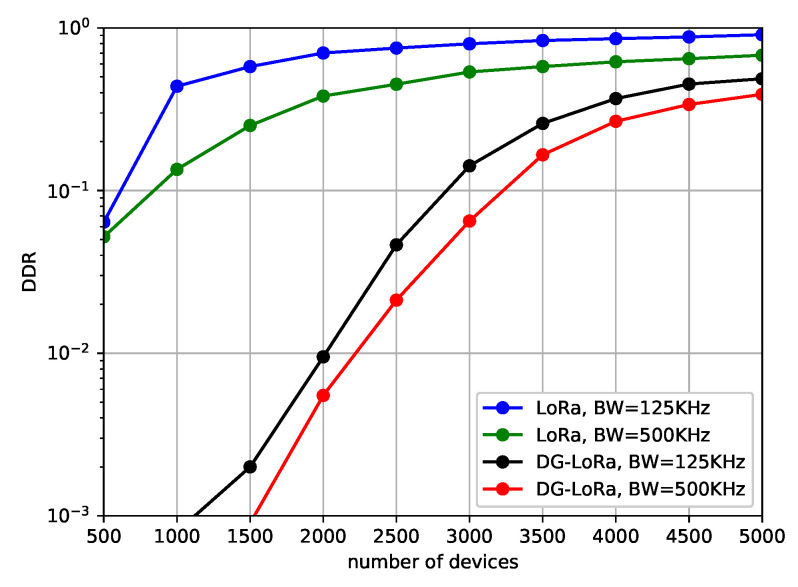
Data drop rate according to the channel bandwidth.

**Figure 8 sensors-21-01444-f008:**
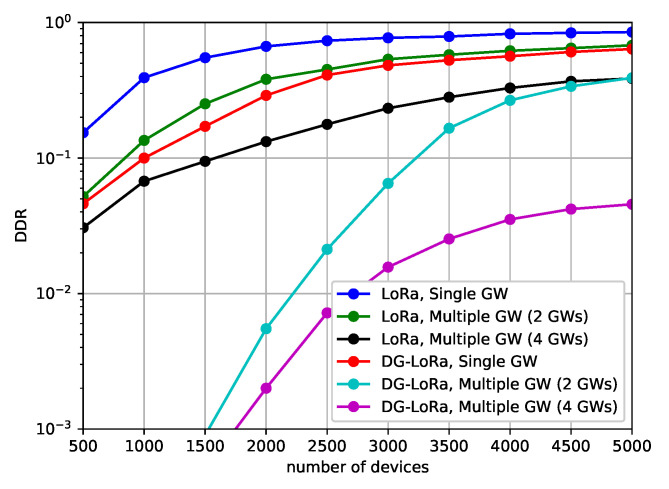
Data drop rate according to the number of gateways.

**Figure 9 sensors-21-01444-f009:**
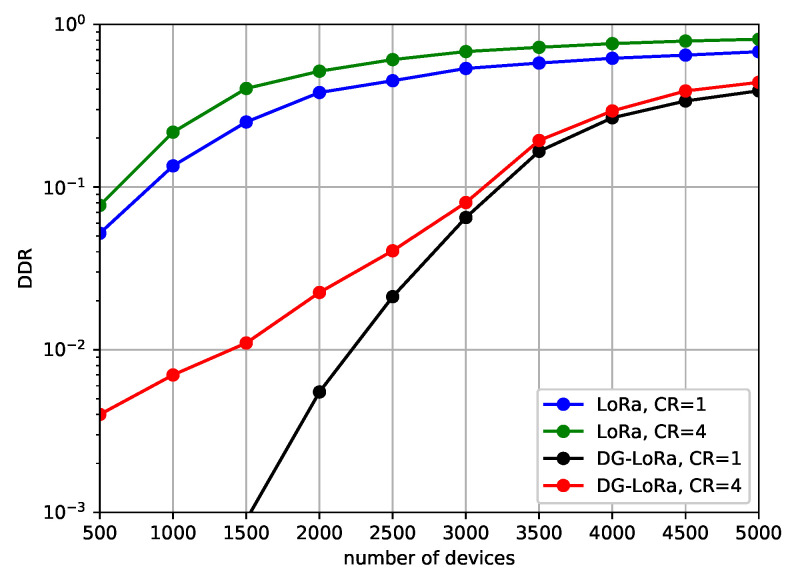
Data drop rate according to the coding rate.

**Figure 10 sensors-21-01444-f010:**
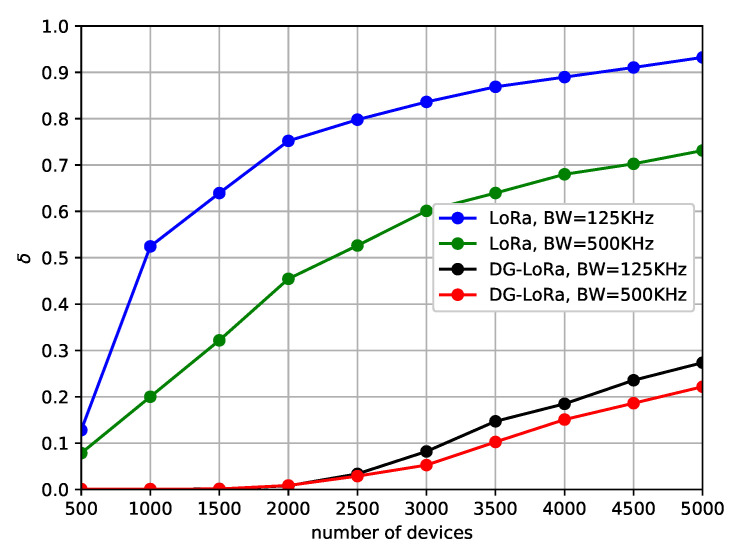
The number of retransmissions according to the channel bandwidth.

**Figure 11 sensors-21-01444-f011:**
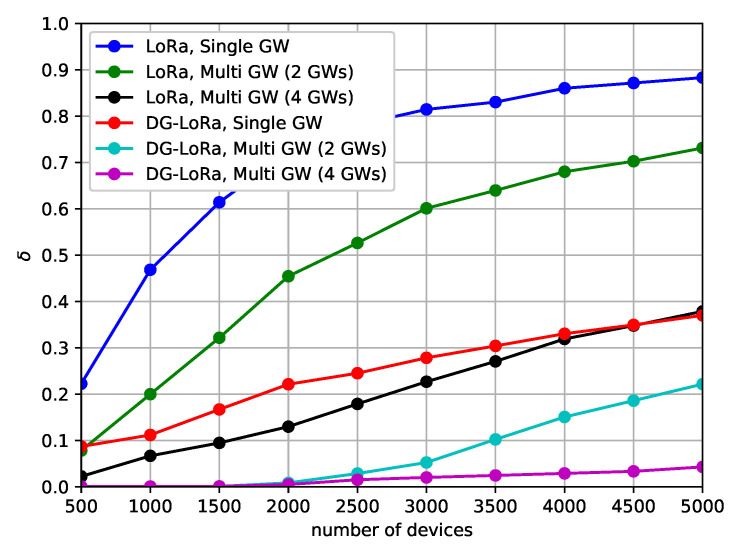
The number of retransmissions according to the number of gateways.

**Figure 12 sensors-21-01444-f012:**
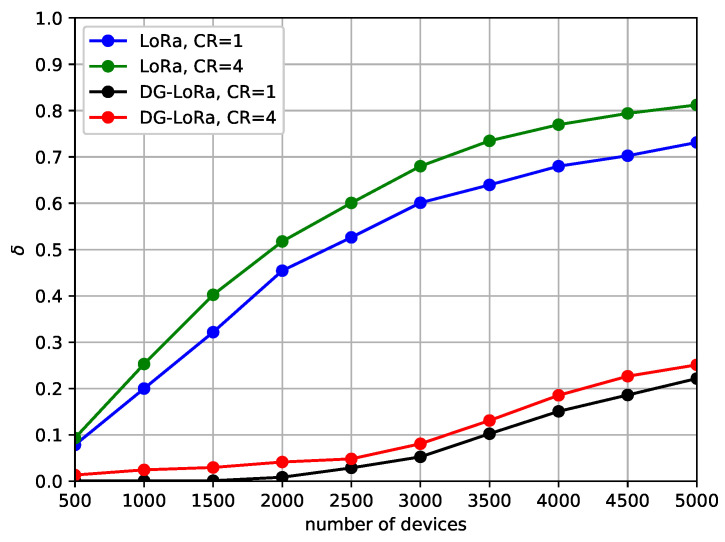
The number of retransmissions according to the coding rate.

**Table 1 sensors-21-01444-t001:** Sensitivity and transmission duration according to spreading factors (SF).

SF	Sensitivity [−dBm]	Duration [ms]
7	−124	36.1
8	−127	72.2
9	−130	144.4
10	−133	288.8

**Table 2 sensors-21-01444-t002:** The maximum number of *gaddr* fields in a GACK according to SF.

SF	Number of *Gaddr* Fields
7	60
8	32
9	13
10	2

**Table 3 sensors-21-01444-t003:** Simulation Parameters.

Parameters	Values
Number of end devices	500–5000
Number of gateways	1–4
Number of subframes	8
Beacon interval	128 s
Number of downlink timeslots	32
MaxNumRetransmission	8
Frame payload	20 bytes
MAC protocol in UTP	unslotted ALOHA
Bandwidth	US 902–928 MHz
RX1DROffset	0
Coding rates	1.4
SFs	7–12
Transmission power	20 dBm
$\bar{L_{p1}\left(l_{0}\right)}$	127.41 dB
γ	2.08
$l_{0}$	40 m
σ	2

## Data Availability

The data presented in this study are available on request from the corresponding author.

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
