# Peer review of "DG-LoRa: Deterministic Group Acknowledgment Transmissions in LoRa Networks for Industrial IoT Applications"

_sensors, 2021, doi:10.3390/s21041444_

Round 1

Reviewer 1 Report

The paper proposes a new MAC protocol called DG-LoRa to improve scalability of LoRa networks. According to the results presented, the protocol provides better performance than LoRaWAN when using multiple end devices. Number of retransmissions is also lowered.

You state that in the beacon period each gateway uses a different SF. How does it know which one to use? Is it pre-defined in the gateway?

Some details in the mechanism (4 and 5.2) could also be more clear. Do you consider that any end device can choose any SF to send data? Or do they use the one received from the gateway in the beacon period? Also, can end devices always connect to any gateway, are they always on range? If that is not the case how can the G4 from Fig.3 respond to any end device if it can only use SF10? Is the SF used by the gateway addressed by the Resource Allocation Algorithm, depending on the devices it has on range (is this what you address in the second paragraph of 5.2)? 

Can you also clarify the notion of finding an idle gateway. How is it "found"? It seems to depend only on the SF that it has used in the previous time slot.

The conclusion that DG-LoRa connects 5 times more end devices while having 90%DDR does not seem to apply to all the tests.

Some minor issues (the line number is the one shown in the pdf, although not all lines have a number):

  • pg.3, line 123: "2SF" or "2SF"?
  • pg.4, line 132: Fig.1 does not show 3 different types of end devices, revise the sentence.
  • pg.5, line 144: before starting a new section an introduction should be provided
  • pg.5, line 146, "beacon period"-> "beacon period (BP)"
  • pg.7, line 196: before starting a new section an introduction should be provided
  • pg.9, algorithm line 2:"there is the idle" -> "there is an idle" (?)

Reviewer 2 Report

The article introduces a new MAC for LoRa in industrial sensor network settings. It introduces group acknowledgements and the authors claim that it increases the number of supported devices by a factor of 5 at the same packet delivery rate while at the same time saving energy.

The topic is of high interest in the community and the claimed results are very nice. However, there are also some weaknesses involved that can be summarized as follows:

  • The energy consumption claims are, in my opinion, not to be taken seriously, as energy consumption is only measured in terms of retransmission counts. Energy that is wasted by, e.g., additional periods of listening the channel is not considered.
  • In particular, it is not clear if the proposed MAC is applicable to a battery powered setup in terms of energy consumption.
  • The simulation setup is not fully described and the presentation of results should be improved.
  • The baseline LoRa scenario is not well-described (e.g. what class? what is the ressource allocation?)

Therefore, I suggest that the authors prepare a major revision of their manusript addressing my comments.

Particular comments to the manuscript:

* Page 6: The number of gaddr fields per GACK of course depends on the spreading factor. Can you include a table indicating this number for SF=7 ... 10 for comparison (example setup)?
* Equation (15): The averaging of the \rho_j only makes sense if each of the devices j\in J has the same amount of transmissions. Otherwise, use a weighted average.
* Chapter 6.2: "Energy is consumed by turning on the RF transceiver for the retransmission of the data frame when the transmission of the data frame fails." Of course it is true that turning on the transceiver for performing retransmissions. However, when comparing two different MAC protocols in terms of energy consumption, it is not sufficient to only count retransmissions. As commonly accepted in the community, a transceiver consumes roughly the same amount of energy for sending, for receiving, and for listening to the channel for a potential message. In particular this "listening" state can vary a lot for different MAC schemes and has to be taken into account by any meaningful comparison of energy consumption. Hence, I suggest that you add such an analysis to your manuscript.
* Already in the abstract it is mentioned that DG-LoRa supports 5 times as many devices at 90% delivery ratio. Do you think that a 90% delivery ratio is of practical interest?
* Related to the previous comment: Figures 7, 8, and 9 show plots of DDR values between 0 and 1. However, not the full range is of the same interest for a real network setup. In fact, if the DDR falls below even 60% it implies that the network is no longer operative. In the end, only the topmost part is interesting, which is why in many publications that plot a delivery rate the y axis is scaled differently, only showing the interesting range. Alternatively, one could plot the outage rate instead (outage rate = 1 - ddr) and use a log scale on the y axis, which would provide a particular emphasis on the very low outage = very high DDR regime.
* Monte-Carlo simulation: What is the area size l used in your simulation? What is the resulting node density for 500 and 5000 nodes?
* Is it possible to apply a similar optimization of the ressource allocation to a regular LoRa MAC without using the GACK? If yes, what would be the performance?
* What is the effect of lost GACK messages on the overall performance and how likely is it in your setup that a GACK message is getting lost?
* In industrial applications it is often required that sensors are battery powered. This implies that energy consumption is the most important metric in such a setup. Typically, Class A LoRa is applied in such settings as the only option to reduce power consumption to an appropriate level by reducing the required listening time to a few ms after a transmission. How does this compare to the DG-LoRa approach in terms of listening time, as the nodes all shall listen for beacons?

Reviewer 3 Report

The authors propose DG-LoRa for improving scalability in low power wide area networks.

The proposed approach is interesting but there are some points that the authors should better discuss.

The authors should be better described the novelties of their approach with respect to existing ones. In particular, the author should discuss limitation and cons of the examined approaches that the proposed one aims to overcome. Furthermore, the authors should provide more details and discussion about the obtained results. The Discussion section also needs to be improved by analyzing the outcome of evaluation section.

I suggest to further analyze more recent approaches about the examined topics. In particular, I suggest the following papers to further investigate predictive maintenance issues in industry 4.0:

1) Comparative study of LPWAN technologies on unlicensed bands for M2M communication in the IoT: Beyond LoRa and LoRaWAN. Procedia Computer Science155, 343-350. 2) Model based vehicular prognostics framework using Big Data architecture. Computers in Industry, 115, 103177

Finally, I suggest to perform a linguistic revision.

Round 2

Reviewer 2 Report

The authors addressed my Comment 2, 3, 4, 5, 7, 8, and 10 satisfactorily and hence these comments are resolved. However, one topic still needs the attention of the authors in my opinion, which is related to the energy consumption of the LoRa nodes.

My particular comments are as follows:

  • Comment 1, 6, 12: Thank you for your extensive and thoughtful replies. However, your replies are not exactly addressing my point, so please let me explain in more detail (and sorry for being unclear in the first review):
    You are right when arguing that listening for a preamble takes a much shorter time than receiving a packet (in particular for high SF). However, in case of LoRa Class A a sensor node would have two slots, i.e. two opportunities to receive after each transmissions. So it would need to listen at these two occasions only, which might be where rarely the case. For some particular LoRa hardware, in case of 1 tx / day you would achieve a battery life time of 5 years regarding specs. When sending once per 10 minutes, the battery lasts a month at most! All this is measured with Class A - I just mention this to put some emphasis on the relevance of the tx, rx and "listening" durations on the battery life time! In the end, if the battery last less than 1 year, it is not feasible in industrial applications.

    Now, if a sensor node would have to be ready to receive packets outside this two slots, it would have to be "listening" to the channel whenever it is possible that such a transmission might arrive, and most of this time is "idle" time, i.e., no transmission is taking place. During this "listening" a significant energy consumption might be assumed. Therefore, the duration of this "listening" time is relevant and not so much the length of the preamble.
    Based on your response to Comment 10, you seem to be aware of all my comments so far, as you state that DG-LoRa is consuming more energy for listening than Class A standard LoRa.
    The critical question is: How much more? It might as well be that you send 10 times a day, but have to listen 30 minutes per day for incoming messages.... Then suddenly the listening becomes the most significant energy consumer... I do not claim that this is the case with your protocol, however it has to be analyzed.

    Summarizing this comment, I would like to see an analysis of the "listening" times of the sensor nodes in the paper, since I feel it is crucial for the energy consumption claims made by the authors.
  • Comment 9: Your comment is fine, but I would like to ask you to write the area sizes and resulting densities in the paper (the calculation of it is not needed).
  • Comment 11: Should go to the paper as well (maybe a refined version).

Furthermore, there are some minor things to correct:

  • Why do you label the y axes of many figures with "DER", while in the text you use "DDR=data drop rate"?
  • Sec. 6.2: Grammar mistake in "We denote normalized the number of retransmissions as $\delta$..."

Overall, I suggest that the authors carefully address my comments and prepare another revised manuscript.

Author Response

We would like to appreciate the reviewer’s careful and valuable comments to improve the manuscript. Attached are the responses to the reviewer’s comments. Please see the attachment.

Reviewer 3 Report

I think that the authors have addressed all my concerns.

Author Response

We would like to appreciate the reviewer's careful and valuable comments to improve the manuscript.